# Seroprevalence of anti-SARS-CoV-2 antibodies in a cohort of New York City metro blood donors using multiple SARS-CoV-2 serological assays: Implications for controlling the epidemic and "Reopening"

Daniel K. Jin[1]☯, Daniel J. Nesbitt[1]☯, Jenny Yang[1], Haidee Chen[1], Julie Horowitz[2], Marcus Jones[2], Rianna Vandergaast[3], Timothy Carey[3], Samantha Reiter[3], Stephen J. Russell[4,5,6], Christos Kyratsous[7], Andrea Hooper[7], Jennifer Hamilton[7], Manuel Ferreira[2], Sarah Deng[7], Donna Straus[8], Aris Baras[2], Christopher D. Hillyer[1,8], Larry L. Luchsinger☉[1]*

1 Laboratory of Stem Cell Regenerative Research, Lindsley F. Kimball Research Institute, New York Blood Center, New York, NY, United States of America, 2 Regeneron Genetics Center, Tarrytown, NY, United States of America, 3 Imanis Life Sciences, Rochester, MN, United States of America, 4 Vyriad, Inc., Rochester, MN, United States of America, 5 Imanis Life Sciences, Rochester, MN, United States of America, 6 Mayo Clinic Department of Molecular Medicine, Rochester, MN, United States of America, 7 Regeneron Pharmaceuticals, Inc., Tarrytown, NY, United States of America, 8 New York Blood Center Enterprises, New York, NY, United States of America

☯ These authors contributed equally to this work.
* lluchsinger@nybc.org

## Abstract

Projections of the stage of the Severe Acute Respiratory Syndrome-Coronavirus-2 (SARS-CoV-2) pandemic and local, regional and national public health policies to limit coronavirus spread as well as "reopen" cities and states, are best informed by serum neutralizing antibody titers measured by reproducible, high throughput, and statically credible antibody (Ab) assays. To date, a myriad of Ab tests, both available and FDA authorized for emergency, has led to confusion rather than insight per se. The present study reports the results of a rapid, point-in-time 1,000-person cohort study using serial blood donors in the New York City metropolitan area (NYC) using multiple serological tests, including enzyme-linked immunosorbent assays (ELISAs) and high throughput serological assays (HTSAs). These were then tested and associated with assays for neutralizing Ab (NAb). Of the 1,000 NYC blood donor samples in late June and early July 2020, 12.1% and 10.9% were seropositive using the Ortho Total Ig and the Abbott IgG HTSA assays, respectively. These serological assays correlated with neutralization activity specific to SARS-CoV-2. The data reported herein suggest that seroconversion in this population occurred in approximately 1 in 8 blood donors from the beginning of the pandemic in NYC (considered March 1, 2020). These findings deviate with an earlier seroprevalence study in NYC showing 13.7% positivity. Collectively however, these data demonstrate that a low number of individuals have serologic evidence of infection during this "first wave" and suggest that the notion of "herd immunity" at rates of ~60% or higher are not near. Furthermore, the data presented herein show that

**Data Availability Statement:** All raw demographic and serology data has been made available as S1 Data.

**Funding:** Funding for this project was provided by the New York Blood Center (NYBC) and Regeneron Pharmaceuticals, Inc (RPI). RPI provided support in the form of salaries for authors JH, MJ, SR, CK, AH, JH, MF and AB. RPI assisted with the design of the study, but did not have a role in data collection and analysis, decision to publish or preparation of the manuscript. The specific roles of these authors are articulated in the 'author contributions' section. There was no additional external funding received for this study.

the nature of the Ab-based immunity is not invariably associated with the development of NAb. While the blood donor population may not mimic precisely the NYC population as a whole, rapid assessment of seroprevalence in this cohort and serial reassessment could aid public health decision making.

## Background

The Severe Acute Respiratory Syndrome Coronavirus (SARS-CoV)-2 pandemic has swept the global community with the United States reporting nearly 8.5 million confirmed cases and over 230,000 deaths from Coronavirus disease (COVID)-19 [1, 2]. Transmission models of SARS-CoV-2, supported by studies of immune responses to related viral infections, suggest that recovery from infection could provide immunity to reinfection [1, 3]. Thus, the use of serological tests to identify those who have acquired antibodies (Abs) against SARS-CoV-2 (seroconversion) and the frequency of seroconversion in the population (seroprevalence) is a powerful means with which to guide public health policies [4, 5]. The term 'hotspots' has emerged to describe regions of high infectivity that appear and then recede as the pandemic evolves. It is important to ascertain the frequency of SARS-CoV-2 seropositivity in regional populations to estimate the risk of infection associated with newly developing or receding COVID-19 hotspots.

As natural infection continues to persist, and vaccine distribution commences, serologic assays will be vital in monitoring the development of herd immunity, also called community or population immunity, which refers to the point at which enough people are sufficiently "protected", and person-to-person transmission is unlikely. Reaching this milestone will, in effect, herald the end of the COVID-19 pandemic. Therefore, population-wide serological assessment and reassessment are critical, and the tests employed need to be reliable, credible, reproducible and high throughput. Furthermore, it is important to understand the degree of correlation of any given assay's "reactivity" with the presence of neutralizing antibody (Nab). These data, then, can be used to assist public health officials in modeling projections and in informing policy making decisions including the safe "reopening" of cities, states, and regions.

The performance and sensitivity of COVID-19 serology assays is myriad in platform (lateral flow, ELISA, etc.) and variable in terms of sensitivity and specificity [6, 7]. Such assays rely on detection and quantification of antibodies that recognize specific SARS-CoV-2 antigens including the four major structural proteins; spike (S) protein (containing the S1 domain and RBD motif), nucleocapsid (NP) protein, membrane (M) protein, and envelop (E) protein [8]. Research conducted on 2005 SARS-CoV-1 and Middle East respiratory syndrome Coronavirus (MERS-CoV), which are highly related to SARS-CoV-2, found that recovered individuals produced the strongest immunogenic antibodies against antigens of the S and N proteins [9]. Thus, the development of serological tests for SARS-CoV-2 antibodies has focused heavily on the detection of antibodies against these viral proteins.

As above, antibody-based tests vary considerably in both technology (platform) and target antigen (design) which led to, in May 2020, the FDA reversing its emergency use authorization (EUA) and approval policies in order to help ensure that reliable tests could be used to accurately measure seroconversion in populations. Some tests have received emergency use authorization but population-wide data are limited, and continuous monitoring is necessary to be of practical importance.

Variability in test characteristics, particularly sensitivity, implies that there may not yet be an *ideal* test design and instrument platform, which can lead to variability and potential bias in

the estimation of the level of immunity in various locales or subpopulations [10, 11]. However, two platforms have been widely cited: 1) in-house enzyme linked immunosorbent assays (ELISA), and 2) high-throughput serological assays (HTSA). ELISAs offer wide flexibility for research laboratories to select virtually any antigenic protein of interest and assay patient sera to provide highly sensitive, quantitative results. HTSAs are more suitable to clinical laboratories processing large volumes of samples. Although HTSAs offer a narrower selection of antigen choices, these platforms offer high-throughput capacity, high sensitivity and can be integrated into clinical lab testing facilities. The resulting expectation of antibody development is an association with antiviral activity and acquisition of immunity against future viral infection. However, only a subset of virus-specific antibodies will be neutralizing, and the levels of SARS-CoV-2-specific neutralizing antibodies necessary to confer protective immunity following infection or vaccination across the population are not known. Thus, studies that evaluate serological test designs are necessary to associate a serological result with a probability of immunity.

New York City (NYC) was one of the first epicenters of the COVID-19 pandemic and possesses the highest case count per capita in the United States from February to June of 2020 [12, 13]. Seroconversion, therefore, is likely to be substantial in a random sampling of NYC residents. Moreover, the true number of COVID-19 cases may be underreported, resulting in inaccurate case estimates (incidence) and morbidity and mortality rates of SARS-CoV-2 [14].

The objectives of the study reported herein were to determine the seroprevalence of anti-SARS-CoV-2 Ab in blood donors in the NYC metro area at a specific point in time four months after the first NY case, as a surrogate for the population as a whole, an indicator of the stage of the epidemic, and as a baseline for future reassessments, using commercially available serology tests, and characterize the Ab responses in ELISAs and a neutralizing antibody assays, allowing us to ultimately inform city, state and nation-wide efforts to mitigate the pandemic and its attendant social and economic strife.

## Results

### Characteristics of the NYC blood donor population

To estimate seroprevalence, 1,000 blood donor plasma samples were collected at each donation centers sequentially between June and July 2020, encompassing regions proximal to NYC, including Long Island, Westchester County and New Jersey and continued until the study collection was complete (**Fig 1A**). To characterize donors demographically, we cross-referenced donor data to the 2010 U.S. Census dataset [15]. Donors ranged in age from 16 to 78 years with a median age of 48 years (95% CI: 46–49 years), which was older than the New York City median age of 35.5 years and deviated from a Gaussian distribution (**Fig 1B**, $r^2 = 0.708$). The donor group also included significantly fewer female donors (38.5%) compared to 52.5% city-wide (**Fig 1C**). Donors that did not respond to ethnicity or reported as 'Other' composed 15.6% of the donors. Among donors that responded, the distribution of donor race/ethnicity was 73% white, 3.6% black, 3.4% multi-race, and 4.4% Asian, compared to an average NYC Metro distribution of 44% white, 25.55% black, 3.99% multi-race, and 12.7% Asian (**Fig 1D**). These data show skewing of blood donors from the NYC demographics in many categories, which is a known characteristic of the blood donor population.

### High throughput serological estimates

To quantify SARS-CoV-2 seroprevalence in donor samples, the Ortho Clinical Diagnostics VITROS Total Ig Test (Ortho) and the Abbott Labs Architect SARS-CoV-2 IgG (Abbott) HTSA assays were used. Results of the Ortho test yielded 121 positive donors while the Abbott

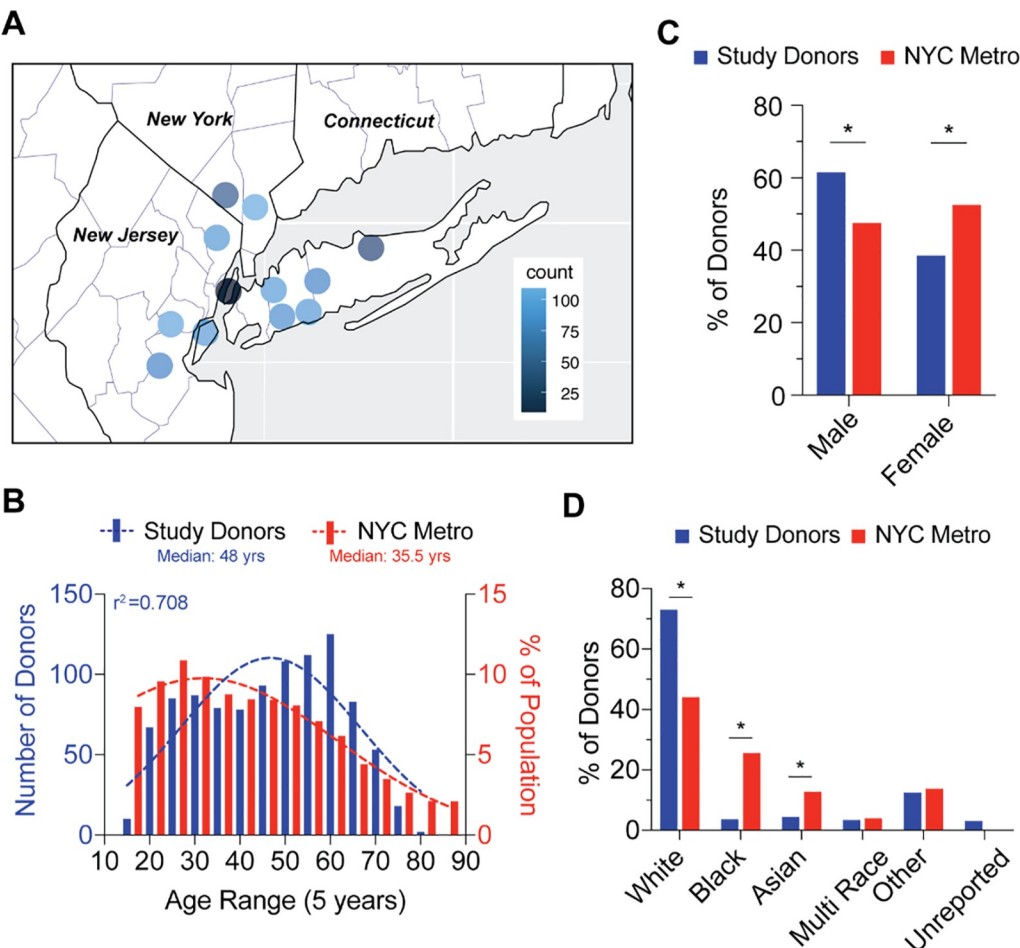

**Fig 1. Blood donor demographics of NYC metro area. A;** Choropleth of donation site locations used for collection of blood donor samples. Heatmap (gradient bar, top) corresponds to frequency of donors collected at each site. **B;** Distribution of NYC Metro area donor age (red bars) compared to NYC demographics (blue bars). Dotted lines represent best fit to a Gaussian distribution and r2 value represents calculated goodness of fit to distribution plot. **C;** Gender frequency of NYC Metro area donors (red bars) compared to NYC demographics (blue bars). Chi-square test for goodness of fit to observed (donors) versus expected (NYC demographics) results; * p<0.01. **D;** Ethnicity frequency of NYC Metro area donors (red bars) compared to NYC demographics (blue bars). Chi-square test for goodness of fit to observed (donors) versus expected (NYC demographics) results; * p<0.01.

test showed 109 positive donors (**Fig 2A**). Adjusting for sample size effect, the estimated seroprevalence rate using the Ortho HTSA was 12.1% (95% CI: 10.2–14.27%) while the Abbott test indicates a seroprevalence rate was 10.9% (95% CI: 9.1% - 12.9%). In total, 128 donors were seropositive by either HTSA test, with 102 donors (79.69%) testing positive for anti-SARS-CoV-2 antibodies using both the Ortho and Abbott tests, and 19 (14.84%) or 7 (5.47%) of donors testing positive using only the Ortho or Abbott test, respectively (**Fig 2B**). The median results using the Ortho test for seropositive donors was 414 (n = 121, 95% CI: 320.0–466.0, IQR: 135.0–692.5), representing a 5,900-fold increase over the median Ortho result for seronegative donors which was 0.07 (n = 879, 95% CI: 0.07–0.07, IQR 0.05–0.10) (**Fig 2C**). The median Abbott test result for seropositive donors was 4.1 (n = 109, 95% CI: 3.56–4.77, IQR: 2.77–5.915), representing a 130-fold increase over the median Abbott result for the seronegative donors which was 0.03 (n = 891, 95%CI: 0.02–0.03, IQR: 0.02–0.5) (**Fig 2D**). We further delineated seroprevalence among sex, age and ethnicity (**Table 1**). We further analyzed HTSA

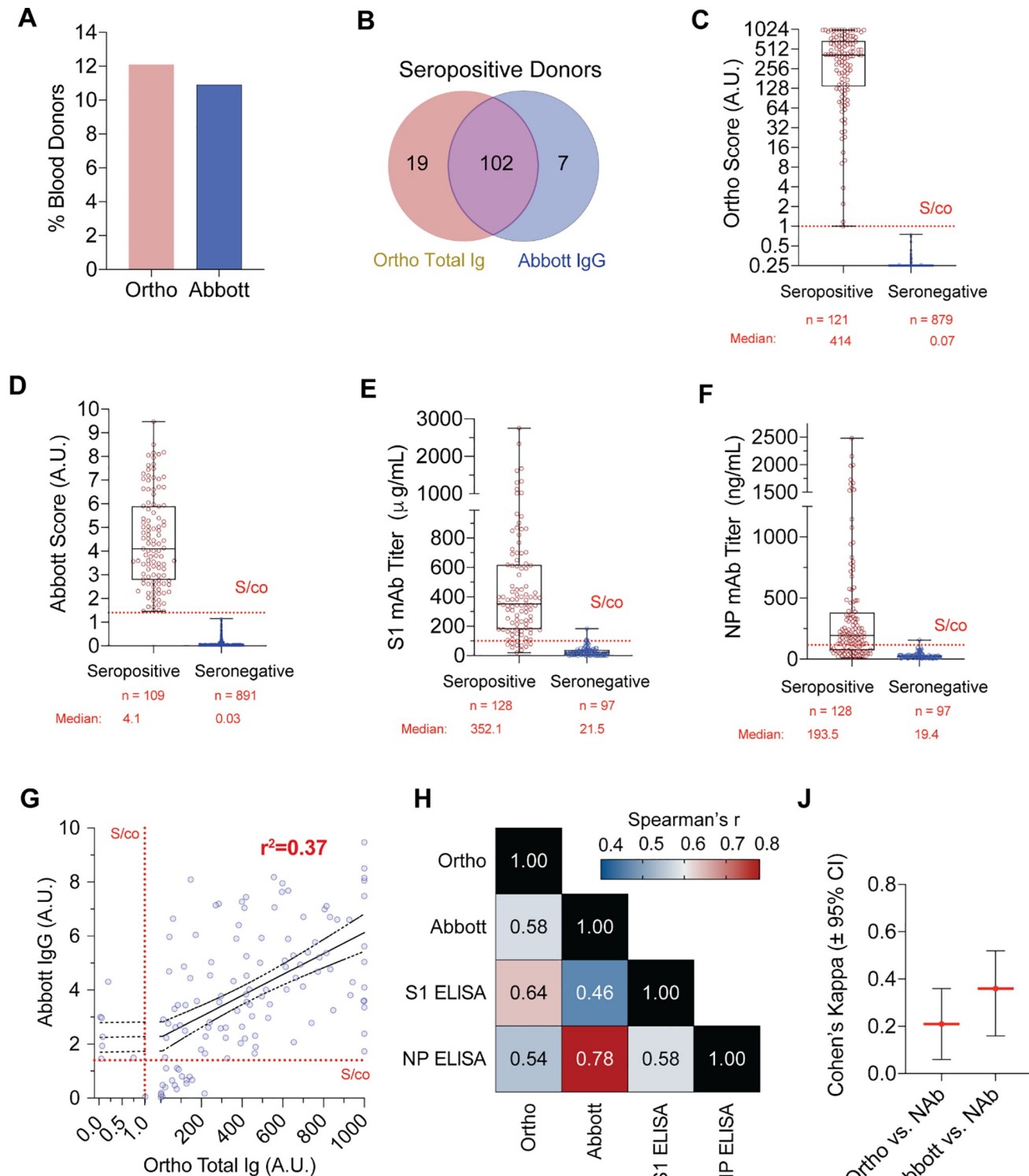

**Fig 2. Serological and neutralizing activity analysis of NYC metro blood donors. A;** Frequency of NYC Metro area seropositive donors as determined using the Ortho Total Ig (yellow bar) or Abbott IgG (blue bar) HTSA assays. **B;** Venn diagram of donors determined to be seropositive using the Ortho (yellow) or Abbott (blue) HTSA assays. Seropositive donors that were reactive for both tests are indicated in overlap (green). **C;** Distribution of Ortho HTSA serological results between seropositive (red dots) and seronegative (blue dots) as determined by the Ortho HTSA assay. Median value and sample number is shown below graph. Dotted line shows S/co value (1.00 A.U.). **D;** Distribution of Abbott HTSA serological results between seropositive (red dots) and seronegative (blue dots) as determined by the Abbott HTSA assay. Median value and sample number is shown below graph. Dotted line shows

S/co value (1.4 A.U.). **E;** Distribution of S1 ELISA serological results between seropositive (red dots) and seronegative (blue dots) as determined by either HTSA assay. Median value and sample number is shown below graph. Dotted line shows S/co value (100μg/mL). **F;** Distribution of NP ELISA serological results between seropositive (red dots) and seronegative (blue dots) as determined by either HTSA assay. Median value and sample number is shown below graph. Dotted line shows S/co value (100 μg/mL). **G;** Linear regression of seropositive donor of HTSA results. Dotted lines denote signal to cutoff (S/co) for each test and goodness of fit, $r^2$, is shown. **H;** Spearman correlation coefficients, r, between each serological assay.

characteristics among demographic categories within seropositive donors. The Ortho HTSA assay showed no significant difference between age groups and the Abbott HTSA assay showed a statistically significant, albeit modest, increase in median scores in donors >55 years versus <30 years (**S1A Fig**, median 3.1 vs 4.4, one-way ANOVA, $p < 0.05$). There was no statistical significance in median scores between sex or ethnicities using either assay (**S1B, S1C Fig**), although the number of donors in these categories is underpowered to draw definitive conclusions. Notably, a higher seroprevalence estimate among females was observed (Ortho, 14.3%) compared to males (Ortho, 10.7%). With respect to age, donors over 65 years had the lowest seroprevalence (Ortho, 6.5% and Abbott, 4.4%). Further, Hispanic/Latino blood donors had higher seroprevalence estimates (Ortho, 14.8% and Abbott, 20.4%) compared to non-Hispanic/Latino donors (Ortho, 10.4% and Abbott, 11.1%), which has been observed in other studies [12]. These data show slightly different seroprevalence estimates, but not serological characteristic, exist between demographic groups in the NYC Metro area.

The gold-standard of serological quantification is the ELISA assay. To compare HTSA results using our in-house SARS-CoV-2 ELISA assays, we analyzed all donor plasma samples that tested positive for either Ortho or Abbott HTSA assays (n = 129) and 100 seronegative for both Ortho and Abbott assays to quantify antibodies against S1 and NP antigens. Using the S1 ELISA (**Fig 2E**), the median value for seropositive donors was 352.1 μg/mL (n = 128, 95% CI: 312.0–399.8 μg/mL, IQR: 179.9–617.2 μg/mL) and the median value for negative donors was 21.5 μg/mL (n = 97, 95% CI: 17.32–26.77 μg/mL, IQR: 6.29–38.79 μg/mL). Using the NP ELISA (**Fig 2F**), the median value for seropositive donors was 193.5 ng/mL (n = 128, 95% CI: 155.6 ng/mL- 226.7ng/mL, IQR: 74.00 ug/mL- 380 ug/mL) and the median value for negative donors was 19.38 ng/mL (n = 97, 95% CI: 15.74 ng/mL—24.20 ng/mL, IQR: 12.79 ug/mL—31.49ug/mL). Interestingly, seropositive donors for Ortho and Abbott tests showed 88.2% and 84.5% above the S/co value for the S1 ELISA assay which demonstrates HTSAs offer an enhanced sensitivity to detect seroconversion. Expectedly, seropositive donors negative by S1 ELISA assays had relatively low HTSA scores (data not shown), which suggests HTSA assays have higher sensitivity than traditional ELISA methodology. Linear regression of Ortho and Abbott tests (**Fig 2G**) showed a modest goodness-of-fit ($r^2 = 0.37$) indicating that while HTSA test scores are positively associated, a high degree of variation within donors exists between HTSA test results. Taken together, these data confirm that a wide range of serological results are prevalent in the NYC metro population and HTSA platforms have the highest sensitivity to quantify serological results with which to estimate seroprevalence.

## Neutralizing activity of NYC blood donors

Antiviral antibodies can inhibit viral particles from infecting target cells and constitute an important form of immunity to future viral exposure; particularly in relation to effective vaccination. In the case of SARS-CoV-2, such assays require biosafety level 3 (BSL-3) facilities and highly trained personnel. To overcome this limitation and expedite testing, we employed a 'surrogate virus' neutralization assay to quantify NAb levels present in donor plasma, which differs from conventional SARS-CoV-2 pseudovirus particles in that surrogate virus retains replication potential and is thus more analogous to live SARS-CoV-2. The results of the

**Table 1. NYC metro seroprevalence estimation within demographic categories.**

| | | | Number Positive | | Seroprevalence Estimates (95% CI, Wilson) | |
|---|---|---|---|---|---|---|
| | | *N* | Abbott | Ortho | Abbott | Ortho |
| **Overall** | | 1000 | 109 | 121 | 10.9 (9.1 to 13.0) | 12.1 (10.2 to 14.3) |
| **Sex** | | | | | | |
| | Men | 615 | 60 | 66 | 9.8 (7.7 to 12.4) | 10.7 (8.5 to 13.4) |
| | Women | 385 | 49 | 55 | 12.7 (9.8 to 16.4) | 14.3 (11.1 to 18.1) |
| **Age** | | | | | | |
| | 18–34 | 281 | 40 | 45 | 14.2 (10.6 to 18.8) | 16.0 (12.2 to 20.8) |
| | 35–64 | 605 | 62 | 70 | 10.2 (8.1 to 12.9) | 11.6 (9.3 to 14.4) |
| | 35–49 | 260 | 25 | 29 | 9.6 (6.6 to 13.8) | 11.2 (7.9 to 15.6) |
| | 50–64 | 345 | 37 | 41 | 10.7 (7.9 to 14.4) | 11.9 (8.9 to 15.7) |
| | 65+ | 114 | 7 | 5 | 6.1 (3.0 to 12.1) | 4.4 (1.9 to 9.9) |
| **Race/Ethnicity** | | | | | | |
| | White | 730 *73.0%* | 64 | 73 | 8.8 (6.9 to 11.0) | 10.0 (8.0 to 12.4) |
| | Black | 36 *3.6%* | 6 | 5 | 16.7 (7.9 to 31.9) | 13.9 (6.1 to 28.7) |
| | Asian | 44 *4.4%* | 8 | 8 | 18.2 (9.5 to 32.0) | 18.2 (9.5 to 32.0) |
| | Multi Race | 34 *3.4%* | 5 | 5 | 14.7 (6.4 to 30.1) | 14.7 (6.4 to 30.1) |
| | Other | 125 *12.5%* | 22 | 28 | 17.6 (11.9 to 25.2) | 22.4 (16.0 to 30.5) |
| | Unreported | 31 *3.1%* | 4 | 2 | 12.9 (5.1 to 28.9) | 6.5 (1.8 to 20.7) |
| | Hispanic/Latino | 108 *10.8%* | 16 | 22 | 14.8 (9.3 to 22.7) | 20.4 (13.9 to 28.9) |
| | Not Hispanic/Latino | 892 *89.2%* | 93 | 99 | 10.4 (8.6 to 12.6) | 11.1 (9.2 to 13.3) |

neutralization end point titer ($NT_{100}$) assays are summarized in **Table 2**. The majority (87.4%, n = 90) of Ortho seropositive donors (n = 121) were positive for Nabs, while 18 samples (14.9%) had indeterminant levels of Nabs and 13 samples (12.6%) were negative for neutralizing activity. Ortho seronegative donors (n = 104) showed 1 positive (0.9%) and 2 (1.8%) indeterminant samples for neutralizing activity. The majority (92.4%, n = 86) of Abbott seropositive donors (n = 109) were also positive for Nabs, with 16 samples (14.7%) having indeterminant levels of Nabs and 7 samples (7.6%) being negative for neutralizing activity. Abbott seronegative donors (n = 116) showed 5 positive (4.3%) and 4 (3.4%) indeterminant samples for neutralizing activity. It was noted that all samples positive for neutralization activity were positive for at least one HSTA assay, while 14.9% of seropositive samples were negative for neutralization activity (**Table 3**). These data illustrate the that serological assays, particularly those with values near the S/co value for each assay, may not reliably correspond to bona fide neutralization activity.

**Table 2. Correlation of serological results with neutralization activity.**

| | Neutralization Result | | | |
|---|---|---|---|---|
| **Ortho HTSA** | **Positive** | **Negative** | **Indeterminate/Borderline** | **Total** |
| *Seropositive* | 90 | 13 | 18 | 121 |
| *Seronegative* | 1 | 89 | 2 | 104 |
| | | | | **225** |
| **Abbott HTSA** | **Positive** | **Negative** | **Indeterminate/Borderline** | **Total** |
| *Seropositive* | 86 | 7 | 16 | 109 |
| *Seronegative* | 5 | 95 | 4 | 116 |
| | | | | **225** |

**Table 3. Neutralization activity as a function of serological results.**

| | Neutralization Result | | |
|---|---|---|---|
| **HTSA Result** | **Positive** | **Indeterminate/Borderline** | **Negative** |
| Ortho Only | 5 | 4 | 10 |
| Abbott Only | 1 | 2 | 4 |
| Double Positive | 85 | 14 | 3 |
| Double Negative | 0 | 0 | 97 |
| Total Samples Tested | 91 | 20 | 114 |
| **Percent HTSA Reactive** | **100%** | **100%** | **14.9%** |

The semi-quantitative $NT_{100}$ method showed that titers for seropositive samples varied between donors. Reciprocal dilution factor values ranged from <80 to 1,280 (**Fig 3A**). Analysis of the 128 seropositive samples revealed 21.9% were below the LOD at the 1:80 dilution (the lowest dilution tested in this analyses) and 7.0% were considered 'indeterminant', due to suspected sample interference. We found low $NT_{100}$ titers of 80 and 160 comprised 30.5% and 29.7% of BD samples, respectively, constituting over half of seropositive blood donors. Moderate $NT_{100}$ titers of 320 and 640 accounted for 5.5% and 4.7% of donors while the highest NAb titers of ≥1280 described 0.8% of seropositive samples. These data indicate that, similar to serology results, NAb levels against SARS-CoV-2 are highly variable and are skewed towards low neutralizing activity within seropositive blood donors in the NYC metro area.

It remains infeasible to implement neutralization assays as a measurement of antiviral antibodies at the scale of the general population. While many serology tests have been developed, evidence as to the predictive value between SARS-CoV-2 serology test results and neutralizing activity continues to be an important validation for the medical and scientific community. To this end, we examined the between serology and neutralization assays in blood donors samples. Spearman's nonparametric correlation analysis showed a high degree of correlation between the various serological assays (**Fig 2H**). As expected, the Ortho test, which measures anti-spike antibodies, showed a higher degree of correlation with the S1 ELISA titers (r = 0.639) while the Abbot test, which measures anti-NP antibodies, showed a high degree of correlation with the NP ELISA titer (r = 0.778). To associate categories of neutralization activity (non-reactive, indeterminant, borderline and reactive) with numerical serological results, we calculated the Cohen Kappa coefficient for each HTSA assay (**Fig 2J**). Analysis of the 128 seropositive donors showed fair association between Ortho and Abbott serology tests and neutralization activity results (Ortho κ = 0.21, Abbott κ = 0.36, κ range 0–1) These data confirm that HTSA assays show correlation with neutralizing activity. Further, median values for both HTSA assays increased with higher neutralizing assay titers (**Fig 3B and 3C**) and this observation was also observed in ELISA assays (**Fig 3D and 3E**). These data highlight the utility of HTSA and ELISA assays to predict neutralization activity of plasma samples.

## Discussion

COVID-19 antibody testing has entered public discourse as an important metric in monitoring the evolution of the SARS-CoV-2 outbreak. Ultimately, the application of antibody testing could be clinically informative as to the degree of immunity incurred by recovered patients or vaccinated individuals. Random blood donor screening is a practice that is readily feasible using blood banking infrastructure to rapidly screen regional populations for seroprevalence monitoring. This is the first study to evaluate a large cohort of random blood donors in the NYC metro area for SARS-CoV-2 antibodies. However, we recognize the limitations of the

current study include a lack of generalizability as a consequence of the modestly skewed demographics of blood donors and the general population as a whole, and that this may impact the conclusions of the results. Thus, more inclusive and complete seroprevalence studies are needed in the future. In fact, seroprevalence has already been suggested to be higher in specific racial/ethnic communities based on recent studies [16]. In particular, Rosenberg et al. measured seroprevalence in 15,000 blood donors by community sampling in grocery stores in New York state during March of 2020 [12]. Hispanics and African Americans represented 17.4% and 13.9% of donors in Rosenberg et al. compared to 10.8% and 3.4% of donors in our study. Blood donor turnout amongst non-white ethnicities has been well documented [17] and although cumulative incidence of SARS-CoV-2 infection and severity has been reported to be higher among non-white groups, underlying mechanisms have not been clearly shown [18]. In another study, Stadlebauer et al. characterized seroconversion from over 10,000 plasma samples from patient groups in the Mount Sinai hospital system using ELISA assays [19]. In the routine care group, seroprevalence increased from 1.6% to 2.2% in March and reached 19.1% seroprevalence by late April 2020. Seroprevalence in the urgent care group reach as high as 67%, likely accounting for the need for medical care associated with an active SARS-CoV-2 infection. Collectively, these studies demonstrate that whole blood donors and community sampling are effective strategies to rapidly surveil immunity within hospital as well as local and state municipalities.

It remains unclear how much neutralizing activity and by extension, antibody levels that are required to prevent reinfection in humans. However, studies designed to test vaccination schedules followed by live-virus reinfection challenge experiments may offer the best laboratory data with which to draw these conclusions. Characterization of the Moderna-1273 vaccine showed that non-human primates on a prime-boost schedule ad mistered a 10ug dose (10% of the human vaccine dose) generated a mean neutralizing titer of 500, while a full 100ug dose generated a mean neutralizing titer of ~3,500 [20]. After a reinfection challenge, the 10ug cohort exhibited protection from reinfection in 7 out of 8 subjects while the 100ug cohort showed full protection as measured by viral RNA detection and both groups showed no sign of pulmonary pathology compared to vehicle controls. Similar neutralizing titers and reinfection challenge results were showed using the Pfizer BioNTech vaccine [21]. These data suggest that only a modest amount of neutralizing activity ($\geq$500) may be required to prevent reinfection and prevent acute respiratory syndrome. Indeed, our analysis of the convalescent plasma donor (CCP) population in NYC found $\geq$50% of CCP donors showed neutralizing antibody activity at or above this threshold. While this interpretation is not definitive proof and will require sophisticated studies to corroborate, existing data supports the conclusion that both natural infection and vaccination can effectively prevent SARS-CoV-2 reinfection.

In this study, we found the Ortho Total Ig and Abbott IgG HTSA assays estimate a ~10.9–12.2% SARS-CoV-2 seroprevalence in July of 2020 in the NYC Metro area. Moreover, we found that ELISA assays, which are the gold-standard of serological quantification, corresponded with seropositive classification of donors as detected by HTSAs, thus validating the use of HTSAs in population studies. In our study, the Ortho HTSA and ELISA assays detected total immunoglobins while the Abbott measured IgG specifically, which could affect seroprevalence estimates. Indeed, we found ~ 24% of CCP donors, who are deferred until they present as asymptomatic for at least 2 weeks, showed detectable IgM using lateral flow assays [22]. Furthermore, differences in the kinetics of anti-S versus anti-NP antibody production and persistence after infection may contribute to serological quantification. This may explain why the Abbott HTSA assay estimated a slightly lower seroprevalence compared to the Ortho HTSA assay in this study. Ideally, the design of seroprevalence estimation studies should adopt assays that measure total immunoglobins to account for variation in Ig production that occurs over

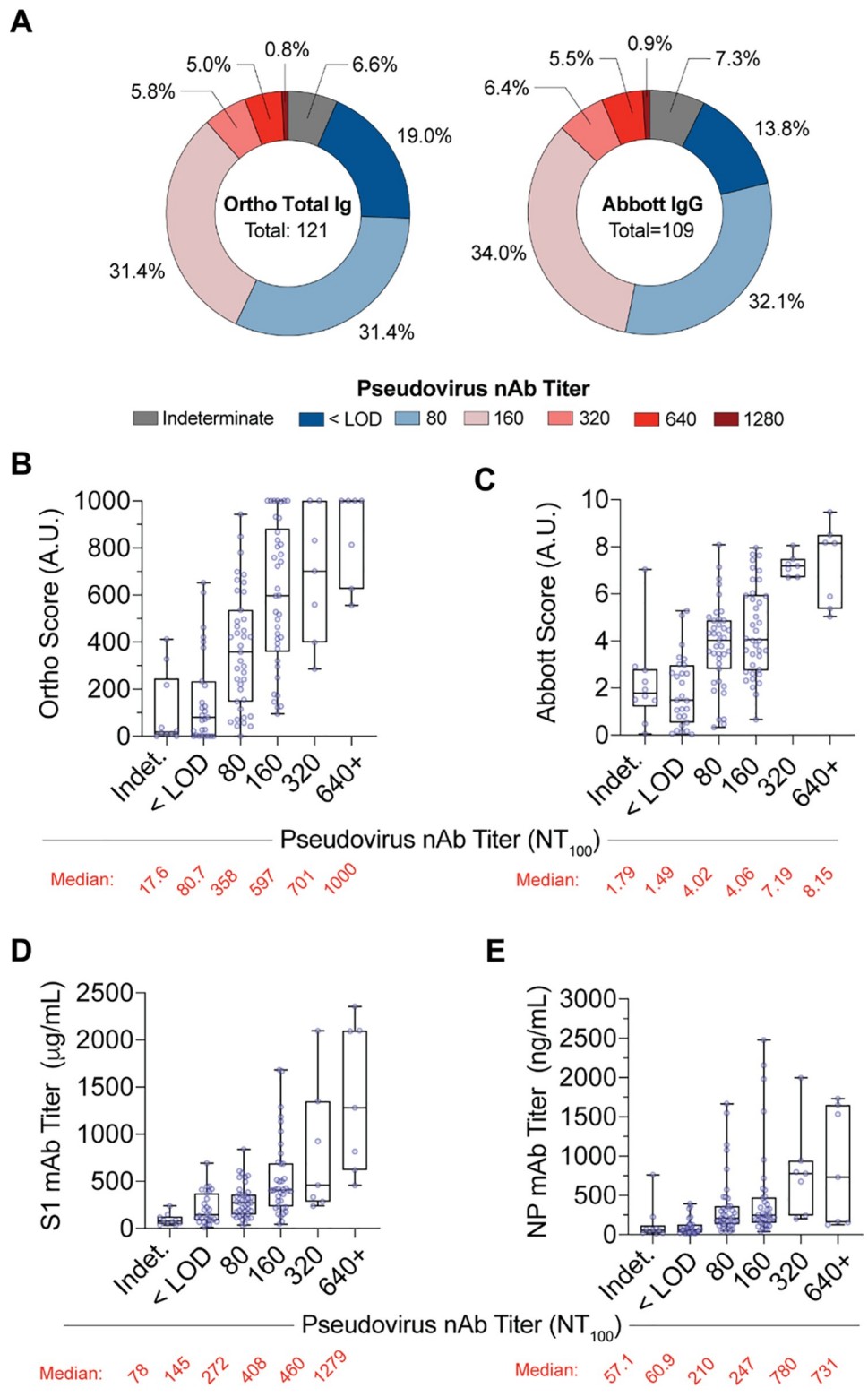

**Fig 3. Correlation of NYC metro donor serological results with neutralization activity. A;** Frequency of Ortho HTSA (left) or Abbott HTSA (right) seropositive donor pseudovirus neutralization end-point titers. **B-E;** Box plots of seropositive donor serology results using the Ortho HTSA, Abbott HTSA, S1 ELISA and NP ELISA for each category of neutralization end point titers. Boxes and whiskers denote 1st and 3rd quartiles and range, respectively. Median serology value of each category is shown below graph.

the course of infection. One limitation to our study is that we could not be certain of infection of the seropositive donors could not be confirmed as diagnostic (PCR) data were not available. Therefore, the Ortho and Abbott assays showed higher sensitivity than ELISA, as stated, but our data could not ascertain specific than the ELISA, although these assays have been validated in other studies. Furthermore, in February 2021, the FDA authorized the use of HTSAs for the quantification of antibodies in convalescent plasma units in an effort to improve the efficacy of passive antibody transfusion therapies [23]. Further, in seropositive blood donors we observed a wide range of anti-SARS-CoV-2-neutralizing activity that was skewed towards low to moderate $NT_{100}$ titers. This trend is in agreement with our previous investigation of convalescent plasma donors [22] and a study of patients recovering from COVID-19, both of which also showed large variability and modest levels of neutralizing activity in plasma units [24].

Our estimation of the NYC Metro area blood donor seroconversion is in agreement with other reports from state and local departments of health. Seroconversion in a study of Bergen County, NJ was estimated to be 12.2% in June of 2020 [25]. Seroconversion among hospital workers in New York City was estimated to be 13.7% as of June of 2020 [26]. The overall seroprevalence in New York City, at the peak of the epidemic, was estimated to be 21% with some communities as high as 68% using data from emergency care clinics [27]. This is juxtaposed to neighboring states, such as Rhode Island, where we estimated seroconversion to be 0.6% among blood donors in May 2020 [28]. Given the early introduction of SARS-CoV-2 in the NYC Metro area in March of 2020 as an initial, and possibly largest, 'hot-spot' in the United States, the estimated seroprevalence in this study may be lower than anticipated due to naturally waning antibody titers [29] (or due to demographics of donor population relative to the NYC population.

## Conclusion

In conclusion, we estimate the seroprevalence of NYC metro blood donors to be approximately 1 in 8 donors during the month of July 2020 and four months post the commencement of the epidemic in NY. While it is slightly lower than another study using a NYC population of healthcare workers during a similar time period, who, in all likelihood, had higher than typical exposure rates [26], our findings demonstrate a comparable seroprevalence estimate can be discerned using a widely accessible blood donor population and it an important metric during this catastrophic outbreak. It is of crucial, albeit underemphasized, importance for public health policy to accurately interpret seroprevalence estimates not only in quantity of persons with immunity, but in quality. In this study, we associated HTSA and ELISA results with neutralizing antibody titers which will be helpful in assessing whether 'heard immunity' is present not only as a proportion of the population, but the degree of neutralizing activity immunity present. Blood donation centers are therefore uniquely suited to be incorporated into future seroprevalence studies to implement rapid seroconversion/seroprevalence monitoring. Furthermore, considering the possibility that this may be an underestimate of the metropolitan population, these conclusions suggest that in the absence of a vaccine, "background" or "herd" immunity continues to be low at four months post-commencement, and, now eight months into the US pandemic, it is probable that the susceptible population remains very high, and possibly at ~80% or greater.

## Methods

### Ethics statement

Approval for donation and collection of blood from donors was attained by written consent. All donors were over 16 years of age. Ethical oversight of seroprevalence studies were obtained from the Institutional Review Board (IRB) of the New York Blood Center.

## Whole blood donors and sample preparation

From June 16, 2020 –July 15, 2020, consecutive NYC metro donors (n = 1,000) received a 2-question survey, provided demographic information and completed a blood donation. The density plot choropleth of donor zipcode prefixes was genereated using ggplot2 in R Studio. Plasma was isolated from whole blood samples collected in citrate tubes. Samples were extracted, aliquoted to minimize freeze-thaw cycles, and stored at -80˚C. Donor blood samples were tested using the Ortho VITROS™ SARS-CoV-2 Total Ig assay, Abbott SARS-CoV-2 IgG assay, in-house ELISAs, and the Vyriad IMMUNO-COV™ neutralization assay as described with some modifications [30]. The IMMUNO-COV assay performed here differed from that which was described in the referenced publication in that: 1) plasma samples were heat-inactivated instead of serum samples, which is necessary due to thermal coagulation and 2) neutralization activity was quantified using neutralization end point titer ($NT_{100}$) method and not a standard curve.

## High-throughput serology assays

Plasma samples were barcoded and dispatched to Rhode Island Blood Center (RIBC). Samples were analyzed using the Abbott SARS-CoV-2 IgG chemiluminescent microparticle immunoassay using the Abbott Architect i2000SR (Abbott Core Laboratories), as well as the VITROS Immunodiagnostic Products Anti-SARS-CoV-2 Total Test using the VITROS 5600 (Ortho Clinical Diagnostics). All assays were performed by trained RIBC employees according to the respective manufacturer standard procedures.

## Virus neutralization assays

Plasma samples were heat-inactivated for 30 min at 56$^{o}$, then clarified by centrifugation for 5 min. at 12,000 x g and assayed using a surrogate virus SARS-CoV-2 neutralization assay. A modified version of the IMMUNO-COV$^{TM}$ assay [30], was used in which each plasma sample was serially diluted and assayed at a total of six dilutions, starting at 1:80. The virus neutralizing titer was determined as the reciprocal of the highest dilution at which the sample was still positive for neutralization based on assay performance relative to a pre-defined calibrator consisting of monoclonal anti-spike antibody.

## In-house SARS-Cov2 binding-antibody ELISAs

Flat-well, nickel-coated 96 well ELISA plates (Thermo Scientific; USA) were coated with 2 ug/mL of recombinant His-tagged S1 spike protein (Antibodies Online, ABIN2650338) or nucleocapsid protein spike protein (Antibodies Online, ABIN2650338) specific to SARS-CoV-2 in resuspension buffer (1% Human Serum Albumin in 0.01% TBST) and incubated in a stationary humidified chamber overnight at 4˚C. On the day of the assay, plates were blocked for 30 min with ELISA blocking buffer (3% W/V non-fat milk in TBST). Standard curves for the S1 assay was generated by using mouse anti-SARS-CoV-2 spike protein monoclonal antibody (clone [3A2], ABIN2452119, Antibodies-Online) as the standard. Anti-SARS-CoV-2 Nucleocapsid mouse monoclonal antibody (clone [7E1B], bsm-41414M, Bioss Antibodies) was used as a standard for nucleocapsid binding assays. Monoclonal antibody standard curves and serial dilutions of donor sera were prepared in assay buffer (1% W/V non-fat milk in TBST) and added to blocked plates in technical duplicate for 1 hour with orbital shaking at room temperature. Plates were then washed three times with TBST and incubated for 1 hour with ELISA assay buffer containing Goat anti-Human IgA, IgG, IgM (Heavy & Light Chain) Antibody-HRP (Cat. No. ABIN100792, Antibodies-Online) and Goat anti-Mouse IgG2b (Heavy Chain)

Antibody-HRP (Cat. No. ABIN376251, Antibodies-Online) at 1:30000 and 1:3000 dilutions, respectively. Plates were then washed three times, developed with Pierce TMB substrate (Thermo Scientific; USA) for approximately 5–7.5 min, and quenched with 3 M HCl. Absorbance readings were collected at 450 nm. Standard curves were constructed in Prism 8.4 (Graphpad Software Inc.) using a Sigmoidal 4PL Non-Linear Regression (curve fit) model.

### Estimated seroprevalence & statistical calculations

For HTSA assays, seroprevalence was estimated using the Wilson Bayesian statistical method [31]. Data and statistical analyses were performed and presented using Prism 8 as indicated. For non-parametric correlation of serological assays, the Spearman r correlation coefficient test was performed using Prism 8. To associate categorical neutralization assays with numerical serological results, the Cohen's Kappa test was performed using SPSS. All donor demographic and serological data used in this study can be found in **S1 Data**.

## Supporting information

**S1 Fig. A;** Distribution of Ortho Total Ig (left) or Abbott IgG (right) HTSA scores among seropositive blood donors by age range groups. N = 129, one-way-ANOVA (Kruskal-Wallace test), $^*$ p < 0.05. **B;** Distribution of Ortho Total Ig (left) or Abbott IgG (right) HTSA scores among seropositive blood donors by sex. N = 129, student's T test (two-tailed). **C;** Distribution of Ortho Total Ig (left) or Abbott IgG (right) HTSA scores among seropositive blood donors by age reported ethnicity. N = 129, one-way-ANOVA (Kruskal-Wallace test). (DOCX)

**S1 Data. Donor demographic and serological assay data used in this study.** (XLSX)

## Acknowledgments

We thank Jill Alberigo, Amanda Brites and Kelly Brightman from Rhode Island Blood Center for their help with performing the Ortho Anti-SARS-CoV-2 Total Test and the Abbott SARS-CoV-2 IgG test. We thank Vijay Nandi for assistance with biostatistical methods.

## Author Contributions

**Conceptualization:** Julie Horowitz, Marcus Jones, Rianna Vandergaast, Christos Kyratsous, Andrea Hooper, Jennifer Hamilton, Manuel Ferreira, Sarah Deng, Aris Baras, Christopher D. Hillyer, Larry L. Luchsinger.

**Data curation:** Daniel K. Jin, Daniel J. Nesbitt, Jenny Yang, Haidee Chen, Marcus Jones, Rianna Vandergaast, Timothy Carey, Andrea Hooper, Jennifer Hamilton, Donna Straus, Larry L. Luchsinger.

**Formal analysis:** Daniel K. Jin, Daniel J. Nesbitt, Jenny Yang, Haidee Chen, Jennifer Hamilton, Donna Straus, Larry L. Luchsinger.

**Funding acquisition:** Aris Baras, Christopher D. Hillyer.

**Investigation:** Daniel K. Jin, Daniel J. Nesbitt, Jenny Yang, Haidee Chen, Julie Horowitz, Rianna Vandergaast, Samantha Reiter, Stephen J. Russell, Andrea Hooper, Jennifer Hamilton.

**Methodology:** Rianna Vandergaast, Timothy Carey, Samantha Reiter, Stephen J. Russell.

**Project administration:** Julie Horowitz, Marcus Jones.

**Supervision:** Julie Horowitz, Marcus Jones, Donna Straus, Aris Baras, Christopher D. Hillyer, Larry L. Luchsinger.

**Writing – original draft:** Daniel K. Jin, Julie Horowitz, Christos Kyratsous, Manuel Ferreira, Sarah Deng, Aris Baras, Christopher D. Hillyer, Larry L. Luchsinger.

**Writing – review & editing:** Larry L. Luchsinger.

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
