## [Decision Letter · Decision Letter 0]

21 Jan 2021

PONE-D-20-38554

Seroprevalence of Anti-SARS-CoV-2 Antibodies in a Cohort of New York City Metro Blood Donors using Multiple SARS-CoV-2 Serological Assays: Implications for Controlling the Epidemic and “Reopening”.

PLOS ONE

Dear Dr. Luchsinger,

Thank you for submitting your manuscript to PLOS ONE. After careful consideration, we feel that it has merit but does not fully meet PLOS ONE’s publication criteria as it currently stands. Therefore, we invite you to submit a revised version of the manuscript that addresses the points raised during the review process.

The two reviewer were in good agreement about the merits of the study and requested only minor changes/clarifications to the text.  Please see their detailed comments below.

We look forward to receiving your revised manuscript.

Kind regards,

Nicholas J Mantis

Academic Editor

PLOS ONE

Journal Requirements:

2. Thank you for submitting the above manuscript to PLOS ONE. During our internal evaluation of the manuscript, we found significant text overlap between your submission and the following preprint of a submission to BMC Infectious Diseases, of which you are an author:

https://www.researchsquare.com/article/rs-76664/v1

We would like to make you aware that copying extracts from other publications or submissions, especially outside the methods section, word-for-word is unacceptable. In addition, the reproduction of text from published reports has implications for the copyright that may apply to the publications.

Please revise the manuscript to rephrase the duplicated text, cite your sources, and provide details as to how the current manuscript advances on the other work submitted to BMC Infectious Diseases. Please note that further consideration is dependent on the submission of a manuscript that addresses these concerns about the overlap in text with published work.

3. Thank you for including the following ethics statement on the submission details page:

'Approval for donation and collection of blood from donors was attained by written consent. All donors were over 16 years of age. Ethical oversight of seroprevalence studies were obtained from the institutional review board of the New York Blood Center.'

Please also include this information in the ethics statement in the Methods section of your manuscript.

5. Thank you for providing the following Funding Statement: 

'Yes. Funds for the collection of 1000 whole blood donors was provided in part by Regeneron Pharmaceuticals.'

i. We note that one or more of the authors is affiliated with the funding organization, indicating the funder may have had some role in the design, data collection, analysis or preparation of your manuscript for publication; in other words, the funder played an indirect role through the participation of the co-authors.

ii. We note that one or more of the authors are employed by commercial companies: Regeneron Genetics Center, Imanis Life Sciences, Vyriad, Inc.

a. If the funding organization did not play a role in the study design, data collection and analysis, decision to publish, or preparation of the manuscript and only provided financial support in the form of authors' salaries and/or research materials, please review your statements relating to the author contributions, and ensure you have specifically and accurately indicated the role(s) that these authors had in your study in the Author Contributions section of the online submission form. Please make any necessary amendments directly within this section of the online submission form. 

Please provide an amended Funding Statement declaring these commercial affiliations and also update your Funding Statement to include the following statement: “The funder provided support in the form of salaries for authors [insert relevant initials], but did not have any additional role in the study design, data collection and analysis, decision to publish, or preparation of the manuscript. The specific roles of these authors are articulated in the ‘author contributions’ section.”

If the funding organization did have an additional role, please state and explain that role within your Funding Statement.

Please ensure the amended statement declares *all* the funding or sources of support (whether external or internal to your organization) received during this study, as detailed online in our guide for authors at http://journals.plos.org/plosone/s/submit-now 

Please also include the statement “There was no additional external funding received for this study.” in your updated Funding Statement.

b. Please also provide an updated Competing Interests Statement declaring these commercial affiliations along with any other relevant declarations relating to employment, consultancy, patents, products in development, or marketed products, etc.  

Within your Competing Interests Statement, please confirm that these commercial affiliations do not alter your adherence to all PLOS ONE policies on sharing data and materials by including the following statement: "This does not alter our adherence to  PLOS ONE policies on sharing data and materials.” (as detailed online in our guide for authors http://journals.plos.org/plosone/s/competing-interests). If this adherence statement is not accurate and  there are restrictions on sharing of data and/or materials, please state these.

Please note that we cannot proceed with consideration of your article until this information has been declared.

6. We note that Figure 1 in your submission contains map images which may be copyrighted.

We require you to either (a) present written permission from the copyright holder to publish these figure specifically under the CC BY 4.0 license, or (b) remove the figure from your submission:

b. If you are unable to obtain permission from the original copyright holder to publish these figure under the CC BY 4.0 license or if the copyright holder’s requirements are incompatible with the CC BY 4.0 license, please either i) remove the figure or ii) supply a replacement figure that complies with the CC BY 4.0 license. Please check copyright information on all replacement figures and update the figure caption with source information. If applicable, please specify in the figure caption text when a figure is similar but not identical to the original image and is therefore for illustrative purposes only.

Reviewers' comments:

Reviewer's Responses to Questions

**Comments to the Author**

1. Is the manuscript technically sound, and do the data support the conclusions?

Reviewer #1: Partly

Reviewer #2: Yes

2. Has the statistical analysis been performed appropriately and rigorously? 

Reviewer #1: No

Reviewer #2: Yes

3. Have the authors made all data underlying the findings in their manuscript fully available?

Reviewer #1: Yes

Reviewer #2: Yes

4. Is the manuscript presented in an intelligible fashion and written in standard English?

Reviewer #1: Yes

Reviewer #2: Yes

5. Review Comments to the Author

Reviewer #1: The authors’ stated objectives were 1) to determine seroprevalence of SARS-CoV-2 antibodies in blood donors in the New York City metropolitan area and 2) characterize the antibody responses using ELISA and neutralization assays. In this manuscript, the authors describe a study in which 1000 plasma samples collected from mid-June to mid-July 2020 from blood donors in the NYC metro area were tested for SARS-CoV-2 antibodies using two commercially available assays and an in-house ELISA. Results from the two commercial assays were compared for concordance and used to calculate seroprevalence estimates. Samples from seropositive donors and a subset of seronegative donors were tested on a quantitative in-house ELISA and on a surrogate virus neutralization assay. The authors report seroprevalence estimates of 10.9% and 12.1% for this period and acknowledge a weakness of the study in that the demographics of the blood donor population differ from those of the NYC population in general. They also report on the correlation between the results of the serology and surrogate neutralization assays. The manuscript is generally well written and provides useful information on seroprevalence and neutralizing titers for the NYC area blood donor population following the COVID-19 surge in this region. Statistical analyses were conducted to support some conclusions, but in some cases statistical support is lacking.

Major comments

1. Clarification is needed regarding the population sampled. The results section heading and line 105 refer to blood donors as does the methods section (line 259). Later in the results (line 185) the authors refer to CP donors, which presumably refers to convalescent plasma donors. CP eligibility requirements are also mentioned on line 207. This brings into question the study population as the general blood donor and convalescent plasma donor populations would be expected to differ substantially with respect to COVID-19 seroprevalence. Please clarify the population sampled for this study.

2. Regarding the sampling method, on line 105 in results, it says samples were randomly selected, but in the methods (line 258) it refers to consecutive NYC metro donors who received a questionnaire and provided demographic information. The sampling description would be improved by addressing the following comments and questions.

a) From the description, it seems more accurate to describe this as a convenience sample rather than random selection.

b) Were the survey and demographic data collection standard practice as part of the intake process for all blood donors or were these implemented specifically for this study? Did any blood donors decline to answer the survey or participate in the study? What questions were asked in the survey?

3. Seroprevalence data on the population overall and broken down by demographic characteristics are presented in Table 1. In the results section, the authors report observing differences between different categories (age, sex, ethnicity). The paper would be strengthened by conducting statistical analysis of these data to determine if statistically significant differences are present between demographic groups.

4. Line 184. What data were used to calculate the correlation between serology and neutralization assays? Were signal/cutoff ratios or just the qualitative results analyzed against neutralizing titer? Clarification should be provided on how this analysis was conducted. In addition, it seems a bit overstated to say that the data confirm a strong correlation between the serology and neutralization assays (line 191). This wording should be softened.

5. The data presented in Tables 4 and 5 are of limited value. PCR results were self-reported and a relatively small number reported being PCR positive. The Tables 4 and 5 refer to correlation of results with PCR positive status but no statistics are presented. On line 209, the statement “these results show strong correlation between positive SARS-CoV-2 PCR test results, seropositivity, and neutralization activity and may be suggestive of longitudinal immunity” is not supported by data and should be removed. The performance characteristics of the Ortho and Abbott assays have been determined and therefore this limited amount of PCR data does little to strengthen the paper. The authors should consider revising or removing this section.

6. In the discussion section, the authors note that blood donor samples offer an accessible sample set for seroprevalence studies during widespread outbreaks, but also note that the demographics may be skewed especially with respect to race and ethnicity. As part of this discussion, the authors should cite the paper by Rosenberg et al 2020 (https://doi.org/10.1016/j.annepidem.2020.06.004) which presents SARS-CoV-2 seroprevalence for NYC and surrounding areas from community-based sampling. The race/ethnicity distribution of the sample population is this study is more closely aligned with the that of the NYC area and would serve as useful comparator for this manuscript.

Additional minor comments

1. Line 92. A time frame should be added to this statement regarding NYC having the highest per capita case count in the U.S. This may no longer be true.

2. Lines 145-146. It is unclear what is meant by the sentence “Interestingly, seropositive donors for Ortho and Abbott tests showed 88.2% and 84.5% above the S/co value for the S1 ELISA assay”. The authors should consider re-phrasing this sentence for clarity.

3. Lines 147-148. The true status of the seropositive donors is not known as diagnostic (PCR) data were not available. Therefore, the Ortho and Abbott assays could be more sensitive, as stated, but they also may be less specific than the ELISA, and this should be noted.

4. Line 149. It should be noted that the Ortho and Abbott assays are approved as qualitative assays and s/co have not been validated for quantitative use.

5. Line 278 - 280. The reference cited for the Immuno-CoV assay should be 23 not 19. The authors should define the criteria for a sample to be considered “still positive”.

6. Line 284. Sources for the antigens used in the ELISA should be provided or referenced. RBD is included among the ELISAs antigens, but no data on RBD were presented. Please explain or remove RBD if it was not used in this analysis.

Reviewer #2: Overall, this is a solidly performed study that offers a snapshot of seroprevalence in a location that was highly impacted by COVID-19. The overall objectives of the study were determine seroprevalence of anti-SARS-CoV-2 antibodies between June and July 2020, in NYC and also to establish a baseline that would help inform mitigation efforts for the pandemic. While the first objective is accomplished, the second should be made clearer and forcefully presented.

Donor Population: The use of blood donors as a study cohort is reasonable and provides a good sample size to estimate seroprevalence during the indicated time periods. However, one concern, especially when linking serology test data to conclusions regarding gender, race or, especially, prior SARS-CoV-2 exposure, is that the nature of voluntary blood donation might skew results. In particular, since there are parallel donation programs specifically focused on COVID-19 convalescent plasma, would a person who, knowing that they had been symptomatic or previously PCR positive and was also inclined to become a donor, choose those programs instead and therefore be underrepresented among non-CCP program donors?

Serology testing: The commercial assays are well-performed and, in combination, provide a strong picture of seroprevalence in the study cohort. However, when comparing or contrasting the data derived from the two tests, the authors may wish to comment as to the impact of using two different result measures (total antibody versus IgG). When only measuring the number of positive individuals, the impact of IgM and/or IgA may not contribute much, especially since the kinetics or duration of these isotypes are unclear, however, the use of two different secondary antibodies may be worth noting. The same comment applies to comparing the HTSA’s to the in-house ELISA (total antibody).

In a similar sense, it may be worth noting that the difference between the HTSA’s may not be sensitivity or abundance of reactivity to Spike versus the N antigen, but also kinetics. Several reports show that antibodies to the N antigen may drop off more quickly than those reactive with the spike and given the uncertain timing between onset of symptoms and blood donation, the conclusions may be impacted by a falling N antigen response.

The authors should better explain the numbers of specimens tested using the in-house ELISA. While the 1000 specimens tested using the HTSAs are clearly stated, the 225 tested by ELISA are not. The authors should describe how that subset of the 1000 specimens were selected, so that the results can be related to the HTSA results. Another concern with the ELISA is the use of mouse monoclonal antibodies and an anti-mouse secondary reagent to set a standard curve for human multi-isotype sera. There are a number of human monoclonal anti-spike and anti-N reagents available which might be preferable for quantifying human sera. Further, in the Materials and Methods description of the ELISAs, the source of the target antigens should be included.

Neutralization testing: The authors do a nice job of providing correlations between the pseudovirus assay and their antibody binding tests. In placing the information in context, it remains unclear as to how much neutralizing antibody is protective, as this impacts the overall goal of informing mitigation strategies. One area which might be discussed is protection in the context of CCP, where recent publications, have attempted to identify levels needed for therapeutic results. Further, given that the Ortho Vitros assay figures prominently in this manuscript, the authors could relate their results to the Ortho threshold for high titered plasma, as indicated in the FDA EUA for CCP.

Discussion: The impact of the study for public health should be made clearer. How exactly will these results help to inform mitigation strategies? Further, it would be helpful to know how the HTSA performance characteristics determined in this study compare to those found in other studies. The authors do note several seroprevalence studies in NYC and NYS over the same time period as evaluated here, but one that they may wish to include and discuss are the large studies on NYC seroprevalence performed by the Mt. Sinai group, and, in particular the paper by Stadlbauer et al.

6. PLOS authors have the option to publish the peer review history of their article (what does this mean?). If published, this will include your full peer review and any attached files.

Reviewer #1: No

Reviewer #2: No

---

## [Author Response · Author response to Decision Letter 0]

20 Mar 2021

**NOTE we cannot find the field in "Additional Information" tab to update our Financial Disclosure, Competing Interests statements in the online portal. They have been updated in the Manuscript according to the editor's instructions.

Editorial Comments

1) Thank you for including your ethics statement on the online submission form: "Approval for donation and collection of blood from donors was attained by written consent. All donors were over 16 years of age. Ethical oversight of seroprevalence studies were obtained from the institutional review board of the New York Blood Center. ". To help ensure that the wording of your manuscript is suitable for publication, would you please also add this statement at the beginning of the Methods section of your manuscript file.

**Thank you for noting this, we have added this as Lines 294-297 of the revised manuscript.

2) Thank you for providing the following Funding Statement: 

'Yes. Funds for the collection of 1000 whole blood donors was provided in part by Regeneron Pharmaceuticals.'

i. We note that one or more of the authors is affiliated with the funding organization, indicating the funder may have had some role in the design, data collection, analysis or preparation of your manuscript for publication; in other words, the funder played an indirect role through the participation of the co-authors.

ii. We note that one or more of the authors are employed by commercial companies: Regeneron Genetics Center, Imanis Life Sciences, Vyriad, Inc.

**Thank you for pointing out these issues. We have updated the Author Contributions, Role of Funding Source and added the Competing Interest Statement to the manuscript.

3) We note that Figure 1 in your submission contains map images which may be copyrighted.

**We have regenerated a new choropleth using the USGS National Map Viewer and replaced Figure 1A. We have explained in the methods section that donor geographical coordinate data was generated from the Geocoordinates website. This should avoid any copywrite infringement. However, we note that Geocoordinates does provide a CC BY-SA 2.0 license (http://www.heatmapper.ca/about/contact/) and states that we are allowed to use their maps in publication provided the reference publication is cited, which we have updated in our manuscript. We would prefer the original figure but would be satisfied with the NSGS choropleth. Please let us know which option satisfies your requirements.

4) We note that you have indicated that data from this study are available upon request. PLOS only allows data to be available upon request if there are legal or ethical restrictions on sharing data publicly. For more information on unacceptable data access restrictions, please see http://journals.plos.org/plosone/s/data-availability#loc-unacceptable-data-access-restrictions. 

**We do not declare any ethical or legal restrictions in sharing this data. We have attached a data table of the raw data used to generate all figures in the manuscript.

---

## [Editor Report · Decision Letter 1]

6 Apr 2021

Seroprevalence of Anti-SARS-CoV-2 Antibodies in a Cohort of New York City Metro Blood Donors using Multiple SARS-CoV-2 Serological Assays: Implications for Controlling the Epidemic and “Reopening”.

PONE-D-20-38554R1

Dear Dr. Luchsinger,

We’re pleased to inform you that your manuscript has been judged scientifically suitable for publication and will be formally accepted for publication once it meets all outstanding technical requirements.

Kind regards,

Nicholas J Mantis

Academic Editor

PLOS ONE
---

## [Editor Report · Acceptance letter]

15 Apr 2021

PONE-D-20-38554R1 

Seroprevalence of Anti-SARS-CoV-2 Antibodies in a Cohort of New York City Metro Blood Donors using Multiple SARS-CoV-2 Serological Assays: Implications for Controlling the Epidemic and “Reopening”. 

Dear Dr. Luchsinger:

I'm pleased to inform you that your manuscript has been deemed suitable for publication in PLOS ONE. Congratulations! Your manuscript is now with our production department. 

Kind regards, 

on behalf of

Dr. Nicholas J Mantis 

Academic Editor

PLOS ONE